# Genetic Diversity of Human Respiratory Syncytial Virus during COVID-19 Pandemic in Yaoundé, Cameroon, 2020–2021

**DOI:** 10.3390/microorganisms12050952

**Published:** 2024-05-08

**Authors:** Moïse Henri Moumbeket Yifomnjou, Gwladys Chavely Monamele, Abdou Fatawou Modiyinji, Mohamadou Njankouo-Ripa, Boyomo Onana, Richard Njouom

**Affiliations:** 1Virology Unit, Centre Pasteur du Cameroun, 451 Rue 2005, Yaoundé P.O. Box 1274, Cameroon; henrimoumbeket92@gmail.com (M.H.M.Y.); monamele.chavely@yahoo.fr (G.C.M.); abfatawou@yahoo.fr (A.F.M.); ripamama@yahoo.fr (M.N.-R.); 2Laboratory of Microbiology, University of Yaoundé I, Yaoundé P.O. Box 812, Cameroon; boyomonana@yahoo.fr

**Keywords:** Cameroon, human respiratory syncytial virus, molecular epidemiology, GA2.3.5, GB5.0.5a, COVID-19 pandemic

## Abstract

Worldwide, human respiratory syncytial virus (HRSV) is a major cause of severe infections of the lower respiratory system, affecting individuals of all ages. This study investigated the genetic variability of HRSV during the COVID-19 outbreak in Yaoundé; nasopharyngeal samples positive for HRSV were collected from different age groups between July 2020 and October 2021. A semi-nested RT-PCR was performed on the second hypervariable region of the G gene of detected HRSV, followed by sequencing and phylogenetic assessment. Throughout the study, 40 (37.7%) of the 106 HRSV-positive samples successfully underwent G-gene amplification. HRSV A and HRSV B co-circulated at rates of 47.5% and 52.5%, respectively. HRSV A clustered in the GA2.3.5 genetic lineage (ON1) and HRSV B clustered in the GB5.0.5a genetic lineage (BA9). Differences in circulating genotypes were observed between pre- and post-pandemic years for HRSV A. Predictions revealed potential N-glycosylation sites at positions 237-318 of HRSV A and positions 228-232-294 of HRSV B. This study reports the molecular epidemiology of HRSV in Cameroon during the COVID-19 pandemic. It describes the exclusive co-circulation of two genetic lineages. These findings highlight the importance of implementing comprehensive molecular surveillance to prevent the unexpected emergence of other diseases.

## 1. Introduction

Worldwide, human respiratory syncytial virus (HRSV), a viral pathogen of the lower respiratory tract, affects people of all ages, particularly those younger than 5 years and older than 65 years [1,2]. Data from a systematic review show that episodes of HRSV-ALRI are responsible for 76,612 deaths, highlighting HRSV as the second leading cause of death from tracheobronchial-tree infection globally [3].

HRSV, which belongs to the genus Orthopneumovirus, family Pneumoviridae, is an enveloped virus with a helical nucleocapsid containing single-stranded, negative-sense ribonucleic acid (RNA) [4]. This viral genome consists of 10 genes with a specific 3′–5′ negative-sense sequence that is NS1, NS2, N, P, M, SH, G, F, M2, and L [5,6]. These ten genes are transcribed into ten mRNAs which have a single open reading frame, apart from the M2 gene, which has two overlapping frames. mRNA is then translated into 11 proteins [7]. Two of these eleven proteins are the major surface glycoproteins. One is the attachment protein, G, the most variable, which ensures the binding of the virion to the target cell surface [8]. The second is the fusion protein (F), which promotes the virus fusion with cell membranes during entry and the fusion of infected cell membranes with surrounding cells to form syncytia [9]. The G protein contains two hypervariable regions (HVR1 and HVR2). HVR2 has the highest degree of divergence, which is frequently used for HRSV genotyping and may represent general gene variability. Based on this genetic variability of HVR2, HRSV has been distinctly classified into two groups: HRSV A and HRSV B [10,11]. These groups have been further subdivided into 14 genotypes (GA1-GA7 [11,12], SAA1 [13], NA1-NA4 [14,15], ON1-ON2 [16,17]) for HRSV A and 20 genotypes (GB1-GB4 [11], SAB1-SAB4 [13], URU1-URU2 [18], BA1-BA10 [19,20]) for HRSV B.

However, the standards for genotype designation have not been agreed upon over time. An additional criterion for defining novel genotypes in HVR2 is the presence of a duplicated segment in the HRSV-A and HRSV-B groups, respectively [16,18,19,20,21]. A recent advance in genotyping HRSV A strains using G-ectodomain phylogenetic analysis was employed and used to reassess historical genotypes by calculating mean p-distances within and between genotypes [22]. Therefore, Goya and co-workers have proposed a unified nomenclature to create a new genotype definition based on mean p-distances and phylogenetic analyses [23]. This would standardize strain names and make it easier to infer viral evolution from surveillance data. As a result, HRSV A is reduced from fourteen to three (GA1-GA3) genotypes, while HRSV B is expanded from twenty to seven (GB1–GB7) genotypes. Two further levels of grouping subgenotypes and lineages have been established within this classification. Lineages are defined as GAX.YZ or GBX.YZ, with X, Y, and Z denoting genotype, subgenotype, and a third-level ascending number, respectively.

In addition, there was no licensed vaccine against HRSV until the approval of two vaccines by the US Food and Drug Administration (FDA). These vaccines are effective in the prevention of acute and severe HRSV lower respiratory tract disease (LRTD) and lower respiratory tract infections (LRTIs) requiring hospitalization. The first, Arexvy, licensed in May 2023, is a pre-fusion F HRSV vaccine with an AS0E1 adjuvant (RSVpreF3 OA) designed for adults aged 60 years and older [24]. The second, Abrysvo, was approved in May 2023 for the prevention of LRTD caused by HRSV in persons aged 60 years and older. In August 2023, it was also approved for use at 32 to 36 weeks’ gestation in pregnant women to prevent LRTD and severe LRTD caused by HRSV in infants from birth to 6 months of age [25]. In addition, several other vaccine-development programs are in various stages of development, some of which may be available soon. Therefore, it is important to provide more data on the genetic variability of HRSV, especially during this period of the COVID-19 pandemic, when the development of mRNA vaccines has been remarkably rapid [26], and thus may support the development of new vaccines. During the pre-COVID pandemic period, a few studies from sub-Saharan Africa investigated the genetic characterization of HRSV [27,28,29,30,31,32,33]. For Cameroon, a first study showed that two genotypes, NA1 and BA9, co-circulated during three consecutive epidemic seasons from 2011 to 2013 [34]. In addition, we recently showed that during the COVID-19 pandemic, HRSV was the third most common respiratory virus (9.5%) after SARS-CoV-2 (severe acute respiratory syndrome coronavirus 2) and influenza virus in patients of all ages in Yaoundé [35]. In this study, we investigated the genetic variability of HRSV strains identified in patients from Cameroon in the course of the COVID-19 pandemic, specifically focusing on the second hypervariable region of the G gene (HVR2).

## 2. Materials and Methods

### 2.1. HRSV Samples

Between July 2020 and October 2021, 1120 nasopharyngeal swabs collected from patients with ILI (influenza-like illness) or SARI (severe acute respiratory infection) were tested for HRSV in Yaoundé. Of these, 40.4% (453/1120) were obtained from influenza surveillance, while 59.6% (667/1120) were from SARS-CoV-2 surveillance [35]. During the same period, a total of 106 nasopharyngeal swabs tested positive for HRSV by real-time PCR. This HRSV detection was performed according to methods developed by the US Center for Disease Control (CDC) and Prevention. A sample was considered positive if the cycle threshold was below the cut-off value of 36. Patients ranged in age from 1 month to 88 years, with a median age of 1 year (IQR: 0–17 years). The majority of patients (61.3%) were aged below 5 years, 28 patients (26.4%) were at least 5 years of age, and 13 patients (12.3%) had no specified age. Of the participants, 47 were male (44.4%), 44 were female (41.5%), and the sex of 15 individuals (14.1%) was not specified.

The Ministry of Public Health in Cameroon and the Centre Regional Human Health Research Ethics Committee (CE No 1302/CRERSHC/2021) examined and approved this study. It was carried out in compliance with the 1975 Declaration of Helsinki [36], as amended in 2013. Prior to usage, the SARS-CoV-2 and influenza samples and database were pseudonymized by the surveillance systems. Every safety measure was implemented to ensure the confidentiality of patient identification. Informed consent was not necessary because our study is a retrospective analysis utilizing samples and a database devoid of personally identifiable information.

### 2.2. RNA Extraction, HRSV G Gene Amplification, and Sequencing

Viral RNA was extracted from HRSV-positive samples according to previously established procedures [35]. The HVR2 region of the HRSV G gene was then amplified using ABG490 sense ((ATGATTWYCAYTTTGAAGTGTTC) and F164 antisense (GTTATGACACTGGTATACCAACC) primers [37,38]. Positions 482–504 of the G-gene sequences of the HRSV/A/England/397/2017 reference sequence (EPI_ISL_412866) and positions 476–498 of the G-gene sequences of the HRSV/B/Australia/VIC-RCH056/2019 reference sequence correspond to the ABG490 primer sequence (ATGATTWYCAYTTTGAAGTGTTC). Targeting positions 164–186 of the F-gene sequences of the reference strain HRSV/A/England/397/2017 are the F164 primer (GTTATGACACTGGTATACCAACC). The SuperScript^®^ III one-step RT-PCR system (Thermo Fisher Scientific, Carlsbad, CA, USA) was used in accordance with the manufacturer’s guidelines. Briefly, 45 μL of the PCR reaction mixture comprising the ABG490 sense and F164 antisense primers was combined with 5 μL of sample RNA extract. Following the manufacturer’s instructions, Taq DNA polymerase (Thermo Fisher Scientific, Carlsbad, CA, USA) was used to carry out a semi-nested amplification procedure. Briefly, a 45 μL PCR reaction mixture including sense primers for HRSV A and B (AG655 (GATCYCAAACCTCAAACCAC) and BG517 (TTYGTTCCCTG-TAGTATATGTG), respectively) and the reverse primer F164 was mixed with 2.5 μL of the product from the first PCR reaction. The G gene of the reference strain HRSV/A/England/397/2017 has positions 640–659 that the AG655 primer corresponds to, while BG517 corresponds to positions 502–524 of HRSV/B/Australia/VIC-RCH056/2019 (EPI_ISL_1653999) (OP975389.1). Using the BigDye Terminator v3.1 cycle sequencing kit (Thermo Fisher Scientific, Foster City, CA, USA) and the Sanger method, bidirectional sequencing was performed on fragments generated using semi-nested PCR. The semi-nested PCR primers were utilized for sequencing on Applied Biosystems 3500 series genetic analyzers.

### 2.3. Sequence Alignment and Phylogenetic Tree Construction

Using the CLC Main Workbench software (version 5.5), consensus sequences were generated and put through similarity searches using BLAST. The MEGA software version 7 Clustal W algorithm was used to align every sequence. Pairwise deletion was used to treat alignment gaps, and the bootstrap method with 1000 replicates was used to calculate the standard errors of the estimations [39,40].

For comparative genomics with reference strains HRSV/A/England/397/2017 (EPI_ISL_412866) for HRSV A and HRSV/B/Australia/VIC-RCH056/2019 (EPI_ISL_1653999) for HRSV B, the Nextstrain web tool [41] was used. The most appropriate nucleotide substitution models (GTR + G) for both HRSVA and HRSVB were selected using PhyML version 3.0 software [42] based on smart model selection (SMS) [43]. The construction of phylogenetic trees was accomplished using the maximum-likelihood (ML) method while MEGA 7 was used to estimate evolutionary distances [40], with GenBank reference sequences included in the analysis. Bootstrap values greater than 70% were reported on the consensus trees, and 1000 replicates were used to assess the robustness of the tree topology.

The prediction of potential N-glycosylated sites (NXT, where X is not proline) and O-glycosylated sites was performed using the NetNGlyc 1.0 and NetOGlyc 4.0 Server, respectively [44,45]. O-glycosylated sites were identified based on a score > 0.5 [46]. We attempted to identify positively and negatively selected sites on the HRV2 fragment using Single-Likelihood Ancestor Counting (SLAC), Mixed-Effects Model of Evolution (MEME), and Fixed-Effects Likelihood (FEL) [47]. At a 1% significance level, selection pressure analysis was realized using the Datamonkey website interface.

Nucleotide sequences obtained in this study were submitted in the GenBank database and assigned accession numbers OQ914331 to OQ914349 for HRSV A, and OQ923606 to OQ923626 for HRSV B. The presentation format of sequences from Cameroon was as follows: CMR/year of detection/laboratory number, with CMR denoting Cameroon.

### 2.4. Amino Acid Analysis

By using MEGA 7 standard genetic coding software, nucleotide sequences were translated to amino acid sequences. The prototype strains of HRSV A (GA2.3.5: JN257693, Ontario-Canada) and HRSV B (GB5.0.5a: KY249660, UK) were compared with amino acid sequences [23]. The second hypervariable region of the G protein’s amino acid variability was visualized using AliView version 1.25 [48].

### 2.5. Genetic Distances

For single sequence alignments, MEGA 7 software was used to estimate the average genetic distance within and between clades using the simple p-distance method, which indicates the proportion of nucleotide sites at which the two compared sequences differ [40]. Alignment gaps were managed by pairwise deletion. Bootstrapping with 1000 replicates was used to determine the standard errors of the estimates.

## 3. Results

### 3.1. HRSV Genotyping

This study included a total of 106 HRSV-positive samples previously identified by real-time PCR [35]. Of these, 40 samples (37.7%)—seven from 2020 and thirty-three from 2021—were successfully amplified by semi-nested RT-PCR and sequenced at the G gene. The mean cycle thresholds of the amplified samples were not significantly different from those of the unamplified samples (22.1 ± 4.7 vs. 34.6 ± 1.9). HRSV B (52.5%) was not more prevalent than HRSV A (47.5%) *p* > 0.05 (Table 1).

Variability was observed in the distribution of HRSV groups between the years of diagnosis. In 2020 and 2021, HRSV A accounted for 28.6% (2/7) and 51.5% (17/33), respectively, while HRSV B accounted for 71.4% (5/7) and 48.5% (16/33), respectively. HRSV A strains clustered in the GA2 genotype, GA2.3 subgenotype, and GA2.3.5 genetic lineage (ON1 strains) (Figure 1), whereas HRSV B strains clustered in the GB5 genotype, GB5.0 subgenotype, and GB5.0.5a genetic lineage (BA9 strains) (Figure 2). The reference sequences used to construct Figure 1 and Figure 2 are available in Appendix A respectively.

Of the 40 successfully amplified and sequenced samples, 75% (30/40) were inpatients and 25% (10/40) were outpatients. More than half of the inpatients (53.3%, 16/30) were infected with the GB5.0.5a genetic lineage (Table 2). However, no significant differences were found between inpatients and outpatients for either subgroup.

### 3.2. Analysis of Deduced Amino Acid Sequences

The HVR2s of the 19 GA2.3.5 strains in this study were aligned to the prototype strain of the GA2.3.5 genetic lineage (JN257693), first detected in 2010 [23]. A total of twenty amino acid substitutions were found, including P215L (16/19 84.21%), T220A (1/19, 5.26%), P222A (1/19, 5.26%), T228I (1/19, 5.26%), P230R (1/19, 5.26%), T235A (1/19, 5.26%), I243T (1/19, 5.26%), I243S (1/19, 5.26%), H258Q (18/19, 94.73%), E262K (1/19, 5.26%), H266L (18/19, 94.73%), Y273H (2/19, 10.52%), L274P (3/19, 15.78%), S277T (18/19, 94.73%), G284S (1/19, 5.26%), E287K (1/19, 5.26%), T288A (1/19, 5.26%), S294P (1/19, 5.26%), L298P (2/19, 10.52%), and Y304H (1/19, 5.26%), which were identified in GA2.3.5 compared to the prototype strain JN257693 (Figure 3).

In addition, the HVR2s of 21 GB5.0.5a strains from the present study were aligned with the prototype strain of the GB5.0.5a genetic lineage (KY249660), which was first detected in 2013 [23]. A total of forty-one amino acid substitutions including T186A (1/21, 4.76%), K191E (1/21, 4.76%), T197N (1/21, 4.76%), P214S (1/21, 4.76%), L217P (14/21, 66.66%), P221L (2/21, 9.52%), K222E (1/21, 4.76%), P229S (2/21, 9.52%), K232N (1/21, 4.76%), K232T (1/21, 4.76%), T234A (1/21, 4.76%), K236T (1/21, 4.76%), T237S (1/21, 4.76%), T237I (1/21, 4.76%), T238S (1/21, 4.76%), E239K (7/21, 33.33%), R240G (1/21, 4.76%), S243R (1/21, 4.76%), V249E (1/21, 4.76%), I252T (8/21, 38.09%), T254P (1/21, 4.76%), K256N (1/21, 4.76%), K256E (1/21, 4.76%), T258A (2/21, 9.52%), R260K (1/21, 4.76%), T264M (1/21, 4.76%), I268T (6/21, 28.57%), A269V (7/21, 33.33%), A269T (1/21, 4.76%), T274P (3/21, 14.28%), S275P (2/21, 9.52%), K276Q (1/21, 4.76%), K276R (1/21, 4.76%), K276E (1/21, 4.76%), H277P (1/21, 4.76%), Y285H (2/21, 9.52%), T288I (21/21, 100%), N294Y (1/21, 4.76%), N294H (2/21, 9.52%), S305Y (10/21, 47.61%), and T310I (21/21, 100%) were identified in the study sequences of the GB5.0.5a genetic lineage compared to the prototype strain KY249660 (Figure 4).

### 3.3. Analysis of Genetic Distance

An intragenotype p-distance of 0.02 (2% divergence) was established as the general cut-off for both subgroups since the mean intragenotype p-distances for GA1 and GB1 were, respectively, 0.010 (SE: 0.004) and 0.037 (SE: 0.008) (Table 3 and Table 4).

For the genotypes of HRSV A, the mean p-distance between genotypes was 0.086 SE: 0.011 (9% divergence), whereas for the genotypes of HRSV B, it was 0.052 SE: 0.009 (5% divergence). Furthermore, the mean p-distance between HRSV A genotypes and study sequences was 0.115 SE: 0.017 (12% divergence), while the mean p-distance between HRSV B genotypes and study sequences was 0.057 SE: 0.011 (6% divergence) (Table 3 and Table 4).

Conversely, the mean p-distance between reference sequences of the GA2.3.5 genetic lineage and study sequences within the same lineage was 0.032 SE: 0.010 (3% divergence), and the mean p-distance between reference sequences of the GB5.0.5a genetic lineage and study sequences within the same lineage was 0.031 SE: 0.008 (3% divergence) (Table 5 and Table 6).

### 3.4. Analysis of Glycosylation Pattern and Selective Pressure

In Figure 3 and Figure 4, possible N-glycosylation sites are shown by red rectangles. In our investigation, every Cameroonian GA2.3.5 strain exhibited two expected N-glycosylation sites. For the prototype strain JN257693, the first site is at position 237, and the second site is at position 318. In contrast, the GB5.0.5a strains of HRSV B in our investigation displayed three expected N-glycosylation sites. The first site was identified in the prototype strain KY249660 at position 228. It seems that this glycosylation site had not been lost, despite the substitution of one sequence, P229S. K232N is the particular alteration responsible for the second predicted N-glycosylation site. It appears that three sequences at position 294 for the third site have lost their glycosylation site as a result of certain mutations. Consequently, at position 294, N294H substitution occurred in two of these three sequences, whereas N294Y substitution occurred in the other.

There were 47 serine and 43 threonine residues in HRSV-A and HRSV-B that were both predicted to be O-glycosylated. Nevertheless, neither the HRSV-A nor HRSV-B strains had any sites indicating positive selection at particular loci in the HVR2 region of the G gene. In a similar vein, no sequences contained any locations that were subject to negative selection.

## 4. Discussion

The HRSV’s G protein is a major antigenic determinant of immune neutralization, with the highest diversity and substitution rates [49]. Viral infectivity, longevity, and the ability to evade the host immune response may be influenced by genetic and antigenic changes brought about by positive selection in the C-terminal region of the second HVR of the HRSV G gene [19]. Furthermore, it has been found that specific HRSV genotypes are linked to a higher severity of the disease [50]. This study aimed therefore to characterize the HVR2 region of the G gene for HRSV sequences obtained from Cameroonian patients in Yaoundé during the SARS-CoV-2 pandemic. This study was conducted beyond one epidemic season, from July 2020 to October 2021. As previously described, in Cameroon, HRSV detection peaked in December 2020 and June 2021 [35]. This temporal distribution of HRSV differs from other Cameroonian research studies reporting viruses other than SARS-CoV-2 [34,51], but it is consistent with that described in The Gambia [52] during the epidemic period. This could be the result of the public health efforts put in place to stop the COVID-19 pandemic having a suppressive effect on HRSV cases, which decreased the total number of cases and threw off previously noted seasonal trends in various nations [53,54]. This temporal distribution can potentially be explained by viral interference, a phenomenon in which the presence of one virus inhibits or delays subsequent infection with another virus of the same type through several mechanisms, including, competition for cellular resources, activation of the host immune system, or direct interference with viral replication processes. The result is altered epidemic dynamics [55].

Due to low viral loads, we were unable to amplify more than half of the HRSV samples (62.3%; 66/106) in this study. This outcome is similar to that reported by other authors [31,56,57]. However, similar to the global trend that was recorded from 2017 to 2018 by Tabor et al. [58] and other studies [59,60,61], the HRSV B group was very slightly predominant over the HRSV A group in our study. All HRSV B viruses in this study were clustered with the GB5 genotype, which mostly has a specific 60-nucleotide duplication in the HVR2 domain. These GB5 genotypes were classified into a subgenotype called GB5.0, belonging to the GB5.0.5a genetic lineage as previously reported **[59,60,62,63]**. Furthermore, a duplication of the same 72 nucleotides of the BA genotype was previously reported for the ON1 genotype, a member of the HRSV A group, in Canada by Eshaghi et al. in 2010 [16], and was also observed for the GA2 genotype in this study. These GA2 genotypes were all classified into the GA2.3 subgenotype and the GA2.3.5 genetic lineage. These genotypes have been identified in the USA, France, Italy, and Senegal in studies published by Goya et al., Coppée et al., Tramuto et al., and Jallow et al., respectively [59,60,63,64,65]. Furthermore, it is important to note that the GB5.0.5a genetic lineage highlighted in this study was already circulating in the Cameroonian population during the pre-pandemic years [34]; hence, there were no identifiable changes in HRSV B since the onset of the COVID-19 pandemic, which could explain the change in viral spread. On the other hand, for HRSV A, this study reports for the first time the circulation of a new genotype, the GA2.3.5 genetic lineage (ON1 strains), within the Cameroonian population, which could explain an increase in the viral spread of 47.5% in this study for this genotype, compared to a previous study by Kenmoe et al. that reported a circulation rate of 17.4%.

Viral glycosylation has a direct impact on intracellular protein trafficking, folding, protein cleavage, and the biological activities of proteins. These elements are all directly or indirectly linked to viral replication. The amount of N-linked glycans removed can vary between N-glycans inside viral glycoproteins and can either boost or decrease the replication of the relevant viruses. In this case, glycosylation frequently plays a significant role in viral pathogenicity, and the profile of glycosylation present in viral proteins can be used to identify certain pathogenic characteristics of the virus. The N-glycosylation sites N237 and N318, previously reported in the HRSV A group, were also identified in the sequences examined in this study. In addition, three additional N-glycosylation sites, N228, N232, and N294, were observed in the HRSV B group sequences reported in this work, with the N232 site attributed to the K232N mutation. The acquisition of this additional N-glycosylation site in the GB5.0.5a genetic lineage described in this study could potentially confer a selective advantage. Studies have shown that variations in the carbohydrate side chain of the G protein can influence genotype antigenicity by either activating or inhibiting the binding of specific antibodies [66]. The selective pressure on HRSV may be related to the molecular evolutionary mechanism of the three C-terminal hypervariable regions of the G protein [67]. The antigenicity of the G gene’s C-terminal hypervariable region, which has several epitopes identified by neutralizing antibodies, may be the primary cause of the molecular selective pressure [68,69]. Positive selection indicates a survival advantage under the selective constraints faced by the viral population [70]. Notably, according to three selection models—SLAC, FEL, and MEME—we did not find amino acid substitutions in this investigation, either under positive or negative selection pressure. This result is in contrast to earlier findings made in Cameroon and around the world before the outbreak [32,34,38,71,72].

The estimation of p-distances for HRSV provides critical information for research, epidemiological surveillance, vaccine development, and the implementation of control measures adapted to the genetic evolution of the virus. In this study, we report an average intragenotype p-distance of 0.010 (SE: 0.004) for GA1 and 0.037 (SE: 0.008) for GB1; therefore, an intragenotype p-distance of 0.02 was established as a general threshold for both subgroups. These results differ from those of Goya et al., who reported a general threshold value of 0.03 (3% divergence) [23]. This discrepancy may be explained by the fact that the calculation of genetic distance depends on the selection of a diverse and numerous range of sequences. In this respect, average p-distances within and between clades may change as additional sequences from older strains become available in future publications. However, the calculated p-distances both between and within sequences of the genetic lineages GA2.3.5 and GB5.0.5a, as well as those of the study, yielded identical values of 0.013 (SE: 0.004), which is below the threshold of 0.03. This supports the hypothesis that the sequences studied do indeed belong to the previously described genetic lineages.

Nevertheless, our study has identified certain limitations that should be acknowledged. First, our sample population consisted exclusively of patients residing in Yaoundé, which may limit the generalizability of our findings to the overall genetic diversity of HRSV at the national level. Therefore, to have a more thorough understanding of the relationship between disease severity and circulating genotypes, it is crucial to extend molecular characterization efforts to other regions. Second, we have achieved significant results from our sequencing endeavors, which have primarily focused the G gene of the HVR2 region in HRSV genomes. However, to gain a more comprehensive understanding of the molecular characteristics of HRSV in Yaoundé, it is imperative to expand genomic sequencing to a wider range of genomic regions.

This study provides insight into the HRSV genotypes circulating in Yaoundé, Cameroon, in the midst of the COVID-19 pandemic. A target for many vaccines under development, the HVR2 G-glycoprotein’s genetic characterization showed that two HRSV genetic lineages (GA2.3.5 and GB5.0.5a) were co-circulating during the study period. Notably, the observed GA2.3.5 genotype (ON1 lineage) differed from the previously circulating genotype, potentially accounting for its increased viral spread. This divergence could be attributed to a reduction in protective immunity within the population due to limited exposure to HRSV as a result of non-pharmaceutical interventions, particularly the introduction of barrier measures during the COVID-19 pandemic. These findings underscore the importance of incorporating comprehensive molecular surveillance for respiratory-associated viruses into the COVID-19 strategy in order to proactively prevent unexpected outbreaks of additional diseases.

## Figures and Tables

**Figure 1 microorganisms-12-00952-f001:**
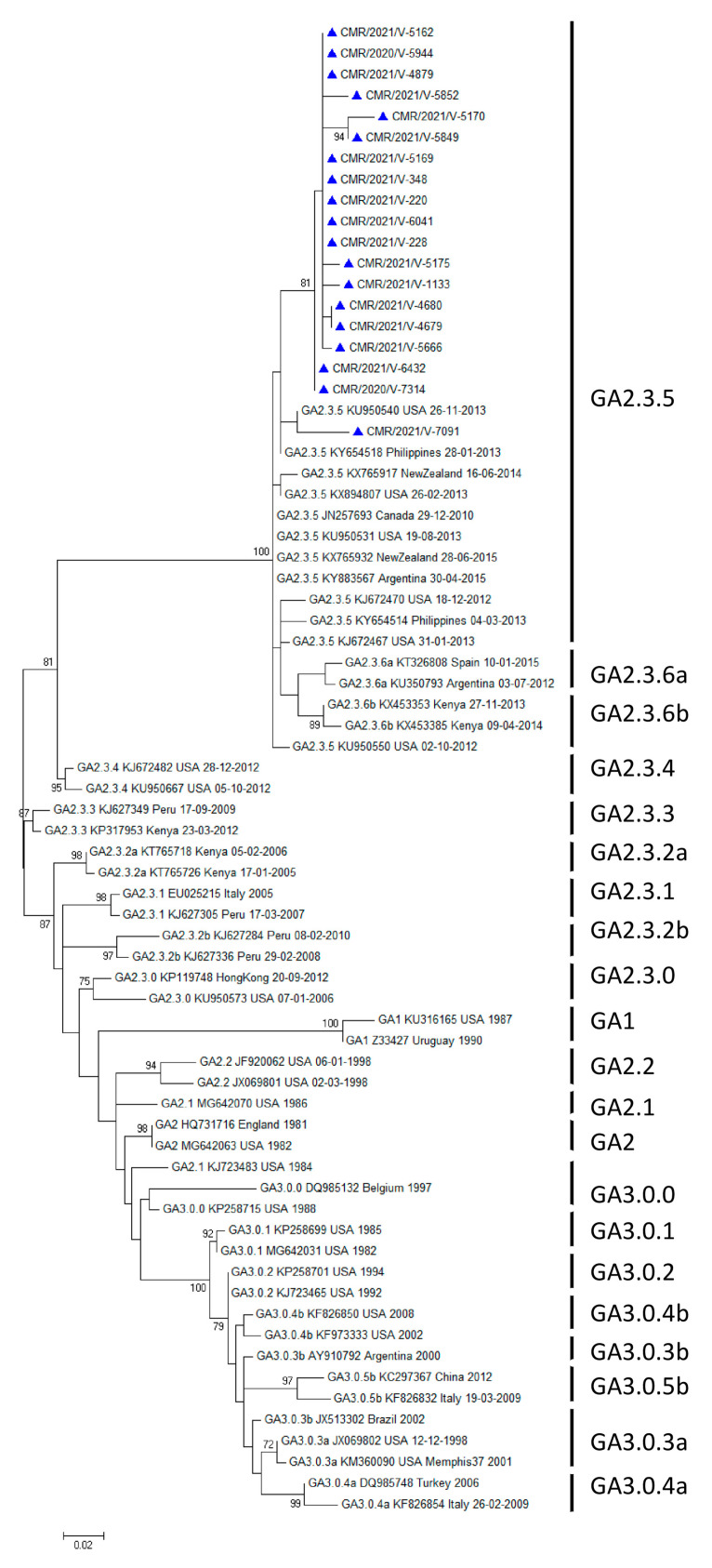
Phylogenetic tree illustrating HRSV A strains identified in Cameroon between 2020 and 2021. Nucleotide sequences from the G gene of the HVR2 C-terminal region were used to build unrooted trees using a Clustal W multiple sequence alignment. The maximum-likelihood method under the General Time Reversible model with the gamma model in MEGA 7 was used. Values at branch nodes indicate the outcomes of bootstrap resampling after 1000 iterations, while scale bars show the frequency of nucleotide substitutions. Shown are only bootstrap values greater than 70%. Genealogical lineages, accession numbers, nations, and years of virus collection are presented from left to right with GenBank reference sequences sourced from several continents. Cameroonian sequences are indicated with a blue triangle.

**Figure 2 microorganisms-12-00952-f002:**
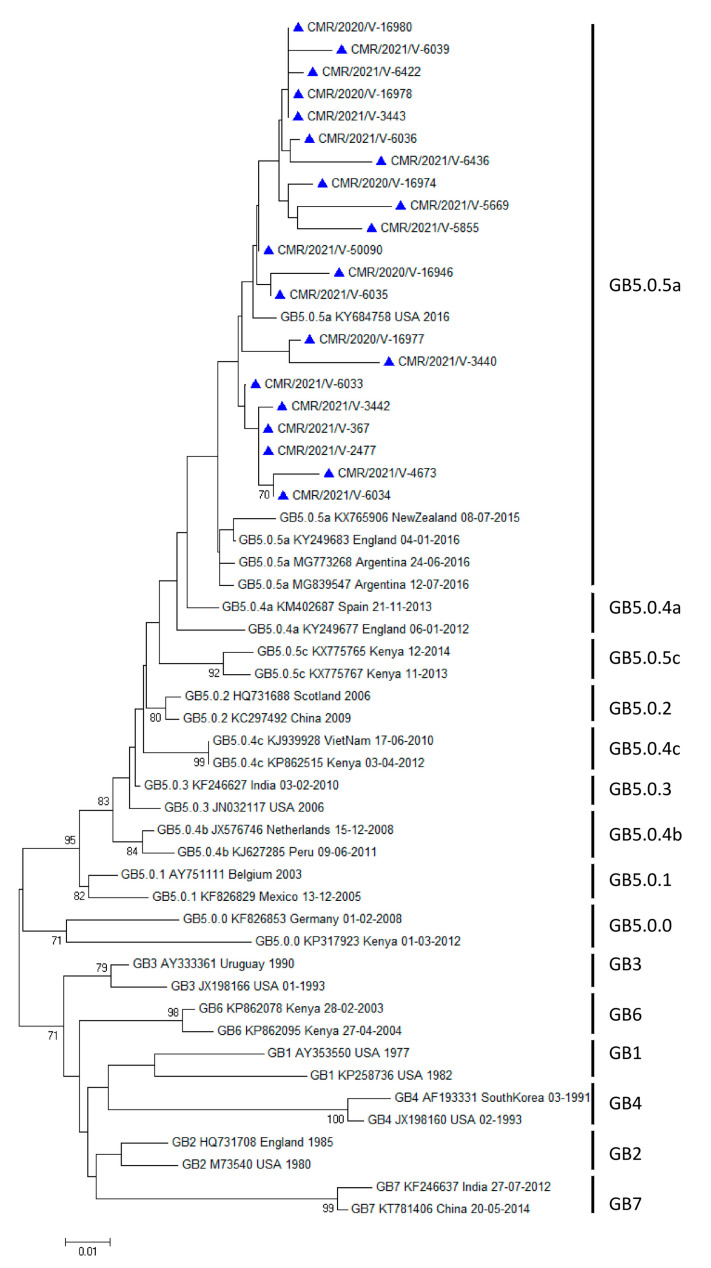
Phylogenetic tree illustrating HRSV B strains identified in Cameroon between 2020 and 2021. Nucleotide sequences from the G gene of the HVR2 C-terminal region were used to build unrooted trees using a Clustal W multiple sequence alignment. The maximum-likelihood method under the General Time Reversible model with the gamma model in MEGA 7 was used. Values at branch nodes indicate the outcomes of bootstrap resampling after 1000 iterations, while scale bars show the frequency of nucleotide substitutions. Shown are only bootstrap values greater than 70%. Genealogical lineages, accession numbers, nations, and years of virus collection are presented from left to right with GenBank reference sequences sourced from several continents. Cameroonian sequences are indicated with a blue triangle.

**Figure 3 microorganisms-12-00952-f003:**
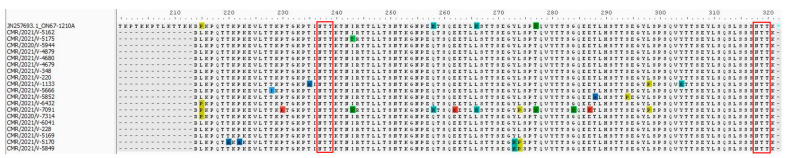
Amino acid alignments from HRSV A sequences in the G gene second hypervariable region. Alignments are displayed in relation to the GA2.3.5 reference strain (JN257693). The alignment matches the G protein’s residues 214 through 321. Dots signify identical residues, while an asterisk denotes the stop codon. Dashes represent amino acids that are missing. Red rectangles represent putative N-linked glycosylation sites (NXT/S, where X is not proline) found with the NetNGlyc 1.0 server.

**Figure 4 microorganisms-12-00952-f004:**
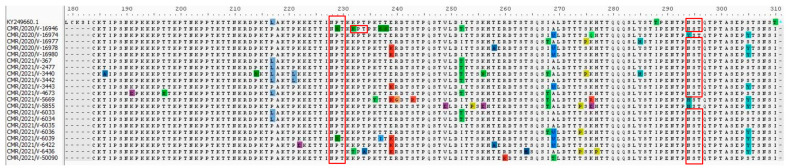
Amino acid alignments from HRSV B sequences in the G-gene second hypervariable region. Alignments are displayed in relation to the GB5.0.5a reference strain (KY249660). The alignment matches the G protein’s residues 184 through 310. Dots signify identical residues, while an asterisk denotes the stop codon. Dashes represent amino acids that are missing. Red rectangles represent putative N-linked glycosylation sites (NXT/S, where X is not proline) found with the NetNGlyc 1.0 server.

**Table 1 microorganisms-12-00952-t001:** Distribution of human respiratory syncytial virus genetic lineages in Yaoundé, Cameroon, 2020–2021.

Genotype	2020 *n* (%)	2021 *n* (%)	Total
HRSV A (GA2.3.5)	2 (10.5)	17 (89.5)	19 (100)
HRSV B (GB5.0.5a)	5 (23.8)	16 (76.2)	21 (100)

*n*: number positive.

**Table 2 microorganisms-12-00952-t002:** Distribution of respiratory syncytial virus genetic lineages by hospitalization status in Yaoundé, Cameroon, 2020–2021.

Genetic Lineage	Total	Inpatients *n* (%)	Outpatients *n* (%)
GA2.3.5	19	14 (73.7)	5 (26.3)
GB5.0.5a	21	16 (76.2)	5 (23.8)

*n*: number positive.

**Table 3 microorganisms-12-00952-t003:** Estimated mean genetic distances between and within HRSV A genotypes and study sequences.

	GA1 (2)	GA2 (34)	GA3 (16)	Study Sequences (19)
GA1 (2)	**0.010** **SE: 0.004**			
GA2 (34)	0.117 SE: 0.012	**0.013** **SE: 0.004**		
GA3 (16)	0.146 SE: 0.013	0.082 SE: 0.011	**0.027** **SE: 0.006**	
Study sequences (19)	0.209 SE: 0.026	0.159 SE: 0.022	0.201 SE: 0.205	**0.012** **SE: 0.005**

SE: Standard error. The number of sequences for each HRSV A genotype and the study sequences are given in brackets. P-distances were calculated both between and among HRSV A genotypes and study sequences. P-distances are bolded within each genotype and study sequence. Additionally, the mean genetic distances are displayed along with SE estimates.

**Table 4 microorganisms-12-00952-t004:** Estimates of mean genetic distances between and within HRSV B genotypes and study sequences.

	GB1 (2)	GB2 (2)	GB3 (2)	GB4 (2)	GB5 (20)	GB6 (3)	Study Sequences (21)
GB1 (2)	**0.037** **SE: 0.008**						
GB2 (2)	0.053 SE: 0.009	**0.024** **SE: 0.006**					
GB3 (2)	0.055 SE: 0.010	0.045 SE: 0.009	**0.015** **SE: 0.005**				
GB4 (2)	0.075 SE: 0.011	0.061 SE: 0.011	0.052 SE: 0.010	**0.013** **SE: 0.007**			
GB5 (20)	0.082 SE: 0.011	0.063 SE: 0.010	0.052 SE: 0.009	0.073 SE: 0.011	**0.034** **SE: 0.007**		
GB6 (3)	0.064 SE: 0.010	0.049 SE: 0.009	0.045 SE: 0.009	0.063 SE: 0.011	0.067 SE: 0.010	**0.027** **SE: 0.006**	
Study sequences (21)	0.099 SE: 0.016	0.082 SE: 0.014	0.080 SE: 0.015	0.113 SE: 0.017	0.050 SE: 0.011	0.092 SE: 0.016	**0.026** **SE: 0.007**

SE: Standard error. The number of sequences for each HRSV B genotype and the study sequences are indicated in brackets. P-distances were computed both between and among HRSV B genotypes and study sequences. P-distances are bolded within each genotype and study sequence. Additionally, the mean genetic distances are displayed along with SE estimates.

**Table 5 microorganisms-12-00952-t005:** Estimates of average genetic distances between and within the GA2.3.5 genetic lineage and study sequences.

	GA2.3.5 (12)	Study Sequences (19)
GA2.3.5 (12)	**0.013** **SE: 0.004**	
Study sequences (19)	0.032 SE: 0.010	**0.012** **SE: 0.005**

SE: Standard error. The number of sequences for each GA2.3.5 genetic lineage and the study sequences are given in brackets. P-distances were calculated both between and among sequences in the GA2.3.5 genetic lineage and those in the study. P-distances are bolded within each genotype and study sequence. Additionally, the mean genetic distances are displayed along with SE estimates.

**Table 6 microorganisms-12-00952-t006:** Estimates of average genetic distances between and within the GB5.0.5a genetic lineage and study sequences.

	GB5.0.5a (5)	Study Sequences (21)
GB5.0.5a (5)	**0.013** **SE: 0.004**	
Study sequences (21)	0.031 SE: 0.008	**0.026** **SE: 0.007**

SE: Standard error. The number of sequences for each GB5.0.5a genetic lineage and the study sequences are given in brackets. P-distances were calculated both between and among sequences in the GB5.0.5a genetic line and those in the study. P-distances are bolded within each genotype and study sequence. Additionally, the mean genetic distances are displayed along with SE estimates.

## Data Availability

The raw data supporting the conclusions of this article will be made available by the authors on request.

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
