# Peer review of "Genetic Diversity of Human Respiratory Syncytial Virus during COVID-19 Pandemic in Yaoundé, Cameroon, 2020–2021"

_microorganisms, 2024, doi:10.3390/microorganisms12050952_

Round 1
Reviewer 1 Report
Comments and Suggestions for Authors
Lines 55-70 are essentially copy-pasted from the Stephanie Goya article, regardless if it is included in the citations.
The 2020 sample is too low to extract any conclusions (that genA reversed the 2020 trend).
Similarly, the conclusions of the study, even if interesting and geographically significant, do not have any effect on future therapeutic approaches for RSV
Finally, nirsevimab is not a vaccine, it is a monoclonal antibody. The second approved vaccine, apart from Arexvy, is Abrysvo
Reviewer 2 Report
Comments and Suggestions for Authors
Dear Editor
Many thanks for asking me to review this original article aimed at investigating the genetic variability of HRSV in Yaoundé during the COVID-19 pandemic.
The paper is well-written and worthy of publication. The authors' experience and the study's prospective design can add significant data to the scientific community.
I have just a few observations:
1. Please improve figure quality and add a figure reflecting the potential underlying mechanisms involved in the viral interference
2. Respiratory viruses can circulate simultaneously and potentially infect the same host, determining different types of interactions, the so-called viral interference. Please add comments in the discussion section focused on viral interference (Refer to Matera L et al. Front Pediatr. 2023 Dec 21;11:1308105).
Comments on the Quality of English Language
Minor editing of English language required
Reviewer 3 Report
Comments and Suggestions for Authors
To Editor, Authors
No doubt, the topic “Genetic diversity of human respiratory syncytial virus during COVID-19 pandemic in Yaoundé, Cameroon, 2020-2021”, Journal of Microorganisms is of interest for virologists, especially for RSV researchers: for epidemiologists of RSV and other ARI viruses during active surveillance.
Results and Discussion are interesting and have some value for medical virologists.
But still has some questions to be addressed.
General Comments:
1. This study was conducted beyond one epidemic season, from July 2020 to October 2021. Authors say this in Discussion , however they separate the years when counting the rates. The main question is few number of samples compared. Since you have too few samples sequenced for 2020, the ratio may be incorrect
2. In Table 2, you could give the percentage for the entire time (2020-2021). at least add a separate column to the table.
3. Also, the distribution of genotypes by year - I think there are too few samples to make such a conclusion. It is necessary to consider at least the entire period. And even in this case, according to your data, an approximately equal ratio is obtained - the differences need to be shown statistically. But there may be too few samples for that. Please address it.
4. If comparing the Results and Discussion sections I see the different number of positives: “This study included a total of 106 HRSV-positive samples. Of these, 40 samples 178 (37.7%) were successfully amplified and sequenced in the G gene, with 7 samples from 179 2020 and 33 from 2021.” “Due to low viral loads, we were unable to amplify more than half of the HRSV samples (62.3%; 66/106) in this study”.
5. Abstract does not reflect clearly what you got. The last paragraph of the discussion section could be more useful as an abstract.
Specific Comments:
6. Chapter 2.1. HRSV samples – details of the real-time PCR testing are missing. Please provide the primers /conditions/or kits details.
7. Line 124. Please check if it is F gene, not G gene: “…targets positions 164-186 of the F gene sequences…”
8. Line 125. Please check “aA semi-nested…”
9. Line 179. Please rephrase “…sequenced in the G gene,…"
10. Line 181 You conclude that HRSV B 181 (52.5%) was more prevalent than HRSV A (47.5%). However there is no statistics, and such conclusion could not be supported. Please check
11. Line 311 – check capital character “of the number…”
12. Line 213 – It is hard to say that the majority of hospitalized patients 53.3% were infected with the GB5.0.5a genetic lineage when we have 16 out of 30.
Reviewer 4 Report
Comments and Suggestions for Authors
The manuscript is logical and consistent, but the work was performed on a very small sample, not sufficient to disseminate the information obtained to the entire population of Cameroon
There are several questions for the authors
1. Introduction
Beyfortus (nirsevimab) is drug, but not vaccine
2. Results
It is not clear how the authors proved that 106 samples were HRSV-positive if only 40 samples could be amplified
3. Table 6
The table content does not match the title
4. The first letter and punctuation need to be corrected
|
aA semi-nested amplification reaction |
125 |
5. Why does a sentence start with a lowercase letter?
|
of the number of serine and threonine residues likely to be O-glycosylated was 47 for |
311 |
|
HRSV-A and 43 for HRSV-B. |
Comments on the Quality of English Language
Minor editing of English language required
Round 2
Reviewer 1 Report
Comments and Suggestions for Authors
The authors should understand that copy-pasting from another article, even when this is referenced, is unacceptable. It was noted in the original review, it has been corrected, but then another part, lines 331-333 is copy-pasted too from the referenced article. Despite the significant improvement of the manuscript, this copy-pasting indicates a disrespect for the reviewers and the future readers.
Reviewer 3 Report
Comments and Suggestions for Authors
I support the study - you significantly improved the text, details and conclusions.
After final grammar checking and correction it could be accepted
